# Impacts of Halogen Substitutions on Bisphenol A Compounds Interaction with Human Serum Albumin: Exploring from Spectroscopic Techniques and Computer Simulations

**DOI:** 10.3390/ijms241713281

**Published:** 2023-08-26

**Authors:** Huan Zhang, Ruirui Cai, Chaolan Chen, Linna Gao, Pei Ding, Lulu Dai, Baozhu Chi

**Affiliations:** 1School of Life Sciences, Nanchang University, Nanchang 330031, China; huanzhang@ncu.edu.cn; 2School of Chemistry and Chemical Engineering, Nanchang University, Nanchang 330031, China; 7803121155@email.ncu.edu.cn (R.C.); 7803121153@email.ncu.edu.cn (C.C.); 7803122101@email.ncu.edu.cn (L.G.); 7803121134@email.ncu.edu.cn (L.D.); 3School of Pharmacy, Nanchang University, Nanchang 330031, China; 4209121042@email.ncu.edu.cn

**Keywords:** halogen-substituted bisphenol A, human serum albumin, spectroscopic techniques, computer simulations

## Abstract

Bisphenol A (BPA) is an endocrine-disrupting compound, and the binding mechanism of BPA with carrier proteins has drawn widespread attention. Halogen substitutions can significantly impact the properties of BPA, resulting in various effects for human health. Here, we selected tetrabromobisphenol A (TBBPA) and tetrachlorobisphenol A (TCBPA) to investigate the interaction between different halogen-substituted BPAs and human serum albumin (HSA). TBBPA/TCBPA spontaneously occupied site I and formed stable binary complexes with HSA. Compared to TCBPA, TBBPA has higher binding affinity to HSA. The effect of different halogen substituents on the negatively charged surface area of BPA was an important reason for the higher binding affinity of TBBPA to HSA compared to TCBPA. Hydrogen bonds and van der Waals forces were crucial in the TCBPA–HSA complex, while the main driving factor for the formation of the TBBPA–HSA complex was hydrophobic interactions. Moreover, the presence of TBBPA/TCBPA changed the secondary structure of HSA. Amino acid residues such as Lys199, Lys195, Phe211, Arg218, His242, Leu481, and Trp214 were found to play crucial roles in the binding process between BPA compounds and HSA. Furthermore, the presence of halogen substituents facilitated the binding of BPA compounds with HSA.

## 1. Introduction

Bisphenol compounds (BPs) are a class of basic chemicals used in the production of polycarbonates and epoxy resins. Among them, BPA is the most used in industry, including in epoxy resins, polycarbonate plastics, dental products, medical devices, electronics, and more [1]. Meanwhile, halogenated derivatives of BPA, such as TBBPA and TCBPA were commonly used as flame retardants [2]. As the commercial applications of BPs have expanded, researchers have increasingly focused on the potential toxicity of BPs, including endocrine-disrupting activity, genetic toxicity, and acute toxicity [3]. Studies have indicated that long-term exposure to BPs may lead to various diseases [4]. Furthermore, BPs can accumulate in the human body through various routes, including skin contact, dietary exposure, and inhalation of dust [5]. Once in the bloodstream, BPs bind to HSA and are transported to tissues and organs through the circulatory system [6]. Hence, the adverse effects of BPs on the body are closely linked to their affinity for binding with HSA. While the impact of BPs on the endocrine system has been extensively studied [7], many mechanisms of BPs, particularly their binding mechanisms with HSA, remain unclear.

With between 55% and 60% of all the proteins in plasma being HSA, it is the most prevalent protein in the plasma of the circulatory system. HSA plays multiple physiological roles, including transportation, pH regulation, and maintenance of blood osmotic pressure [8]. Crucially, HSA is important for the distribution of small molecules [9,10], and it serves as a storage and transport system for both exogenous and endogenous small molecules in the body, including fatty acids, hormones, drugs, amino acids, and proteins [11,12]. HSA contains three similar domains (I, II, and III), each of which are further divided into two subdomains (IA, IB, IIA, IIB, IIIA, and IIIB) [13,14]. Overall, there are seven binding sites in HSA, and the two important binding sites are Sudlow’s sites, namely site I and site II, located in the IIA and IIIA subdomains, respectively [15,16]. Even though these domains have similarities in their amino acid sequence and structure, each domain exhibits different ligand binding affinities and functions. For example, Sudlow’s I preferentially binds with bulky heterocyclic anions such as warfarin, whereas Sudlow’s II has a preferential binding affinity for aromatic carboxylates such as ibuprofen [17]. As the primary reservoir for small molecules in the blood, HSA can bind with ligands in a dissociated state, forming ligand–HSA complexes. The affinity between HSA and ligands is usually expressed by the binding constant (*K_a_*), which represents an important factor influencing the metabolism, distribution, and free concentration of ligands [18], and it significantly affects the half-life of ligands in the human body. A higher *K_a_* value indicates lower tissue absorption of the ligand and a longer elimination half-life in the body [19]. Moreover, the binding of ligands to HSA can change the conformation of HSA, leading to the disruption of its biological function [20,21]. Additionally, toxic small molecules can also be transported, and they are released to target organs with the help of HSA [8,22]. Therefore, the *K_a_* between ligands and HSA can serve as a useful parameter for predicting the potential toxicity of compounds and evaluating their toxicity.

To date, there has been limited research on the interaction between BPA and its analogues with serum albumin molecules. Xie et al. found that hydrophobic interactions were present in the BPA–HSA interaction [23]. Through hydrophobic interactions, BPA could accumulate in the aromatic groups of BSA and adjacent DNA base pairs [24]. Furthermore, Mathew et al. reported the interaction between BPS/BPA with serum albumin and analyzed whether BPS could be a potential replacement for BPA [25]. These studies not only provide new insights into the non-covalent interactions between BPA-like compounds with biomacromolecules but also help elucidate the toxic mechanisms of these harmful chemicals.

However, the mechanism of interaction between halogen-substituted BPA compounds and HSA has not been thoroughly investigated. Therefore, we selected different halogen-substituted BPAs (TBBPA and TCBPA) as representative compounds to thoroughly explore and analyze the binding characteristics of TBBPA/TCBPA with HSA using various spectroscopic techniques and computer simulations. Furthermore, the obtained data combined with theoretical calculation were used to investigate the specific impact of various halogen substitutions on the interaction between HSA and BPA. The results of this research will provide valuable data support for better understanding the distribution, transport, and potential threats to human health posed by different BPA compounds in the human body.

## 2. Results and Discussion

### 2.1. Formation of Complexes

UV-visible absorption spectroscopy is widely used to determine the formation of a ligand–receptor complex [26,27]. The formation of a complex often leads to significant changes in the absorption intensity or maximum absorption peak position of the receptor’s UV-visible absorption spectrum [11]. As shown in Figure 1, HSA exhibits two characteristic absorption peaks at 215 nm and 280 nm, respectively. The peak at 215 nm originates from the n → π* transition of the C=O bond in the peptide backbone of HSA, while another absorption peak corresponds to the information of aromatic amino acid residues in the HSA structure [28].

Upon increasing the concentration of TBBPA/TCBPA, a notable decrease in the intensity of the characteristic absorption peak at 215 nm of HSA is observed, accompanied by a redshift. This observation suggests that TBBPA/TCBPA can bind to HSA, and this bind process change is conformational in the peptide chain of HSA [29,30]. Furthermore, a slight decrease is observed in absorbance at 280 nm, indicating the exposure of aromatic amino acids to a more polar environment [31]. Although the shape of the maximum peak at 280 nm remains largely unchanged, these findings revealed HSA can interact with TBBPA/TCBPA, leading to the formation of TBBPA/TCBPA–HSA complexes.

### 2.2. Interaction Study

Fluorescence spectroscopy is a valuable tool for exploring the binding characteristics between a ligand and receptor [32]. The primary source of HSA’s intrinsic fluorescence is aromatic amino acids, including tryptophan (Trp) residues, phenylalanine (Phe), and tyrosine (Tyr) [33,34]. Among them, Phe is often disregarded due to its poor quantum yield, and therefore, the fluorescence intensity mostly reflects the contributions of Trp and Tyr residues [30]. These residues are located in the binding cavity of HSA, and their fluorescence intensities are sensitive to the presence of ligands. Consequently, fluorescence spectroscopy is well suited for investigating the interactions between ligands and HSA.

Figure 2 depicts the effect of TBBPA/TCBPA on the fluorescence spectra of HSA at 298 K. The maximum emission wavelength of HSA is observed at 342 nm, and its fluorescence intensity systematically decreases as the concentrations of TBBPA/TCBPA increase, accompanied by a slight blue shift in the maximum wavelength. These observations indicate the presence of interactions between TBBPA/TCBPA and HSA, and the microenvironment surrounding Tyr and Trp residues were altered by ligands [11,28]. These results confirm the formation of complexes between TBBPA/TCBPA and HSA, which is in line with the findings from the UV-visible absorption spectroscopy experiments. Moreover, it is noteworthy that TBBPA exhibits a stronger fluorescence-quenching ability compared to TCBPA, indicating a higher binding affinity between TBBPA and HSA.

Fluorescence quenching may occur by dynamic quenching, static quenching, or mixed quenching [35]. To investigate the detailed quenching mechanism between HSA and TBBPA/TCBPA, Equations (1) and (2) [32] can be utilized. If the value of *K_q_* is greater than 2.0 × 10^10^ M^−1^·s^−1^ (the maximum scattering collision quenching rate constant for dynamic quenching), it indicates static quenching [28]. Conversely, if the value is below this threshold, it indicates dynamic quenching. It should be noted that the values of *K_q_* for TBBPA/TCBPA with HSA are significantly larger than 2.0 × 10^10^ M^−1^·s^−1^, revealing that the fluorescence quenching between TBBPA/TCBPA and HSA corresponds to the static quenching mechanism (Table 1).

In addition to *K_q_*, the trend of *K_sv_* with changes in temperature is another important parameter for distinguishing the source of fluorescence quenching [36]. Table 1 presents the *K_sv_* values for TBBPA/TCBPA–HSA under different temperatures. In the TBBPA group, the *K_sv_* values decline with increasing temperature, providing further evidence for the static quenching mechanism. Conversely, the *K_sv_* value for TCBPA–HSA increases with temperature, indicating the occurrence of dynamic quenching in the binding process between TCBPA and HSA [30]. In summary, TBBPA–HSA exhibits a static quenching mechanism, while TCBPA–HSA exhibits a mixed quenching mechanism.

Through thermodynamic experiments, it is possible to comprehend the interaction between TBBPA/TCBPA and HSA better. *K_a_* and n were calculated using Equation (3) and are presented in Table 2. At any temperature, the value of n tends to be close to 1, showing a stable binary complex was formed between TBBPA/TCBPA and HSA. In vivo, binding constants for drug–protein interactions typically fall within the range of 10^4^~10^6^ M^−1^ [20]. Binding constants in the range of 10^4^ M^−1^ and 10^5^~10^6^ M^−1^ generally signify moderate and strong binding strengths, respectively, between the ligand and HSA [37]. The *K_a_* values for TBBPA–HSA and TCBPA–HSA were 6.245 × 10^5^ and 2.040 × 10^5^ M^−1^ at 298 K, respectively, indicating the strong binding affinity in both cases. Notably, the *K_a_* value of TBBPA–HSA is three times higher than that of TCBPA–HSA, suggesting a higher binding affinity of TBBPA to HSA.

In general, ligands primarily exist in either the bound state, where they are mainly bound to plasma proteins, or the free state in the bloodstream [38]. The bound ligand serves as a reservoir or storage, while the free ligand exerts its toxicological and/or pharmacological effects [39]. Hence, the *K_a_* of ligand–HSA was considered to be a useful parameter to access the toxicity of similar compounds [35,37]. For example, the toxicity of aromatic organophosphate flame retardants was inversely proportional to their binding affinity with HSA [35]. The *K_a_* values of TBBPA–HSA are significantly higher than those of TCBPA at different temperatures, indicating a greater stability of TBBPA–HSA compared to TCBPA–HSA. Consequently, the proportion of free TBBPA in the bloodstream is lower than that of TCBPA under the same intake conditions. This suggests that TCBPA may have a higher toxicity in the body compared to TBBPA [40,41]. Moreover, stronger binding results in a longer half-life and slower metabolism in the body, potentially enhancing toxic side effects [42]. Therefore, TBBPA is more likely to accumulate in the plasma with a longer elimination half-life compared to TCBPA. Overall, the long-term effects of TBBPA on HSA should be taken into consideration. These findings emphasize the importance of not overlooking the impact of TBBPA/TCBPA on human health.

To obtain more understanding of the binding forces between TBBPA/TCBPA and HSA, the Δ*G*, Δ*S*, and Δ*H* between HSA and TBBPA/TCBPA were calculated using Equation (4). A negative value of Δ*G* indicates the spontaneous formation of TBBPA/TCBPA-HSA complexes (Table 2) [43]. The values of Δ*H* and Δ*S* provide information about the binding forces between macromolecules and ligands [44,45]. For TBBPA–HSA, the Δ*H* is 19.19 kJ·moL^−1^, and the Δ*S* is 175.4 J·moL^−1^·K^−1^, indicating hydrophobic interactions being the main driving force [46]. Conversely, for TCBPA–HSA, the Δ*H* is −55.35 kJ·moL^−1^, and the Δ*S* is −84.86 J·moL^−1^·K^−1^. Therefore, hydrogen bonds and van der Waals forces play significant roles in the formation of TCBPA–HSA complexes [18,47].

To further analyze the differences in the binding ability between TBBPA and TCBPA with HSA, we employed Multiwfn 3.8 software [48] to calculate and analyze the molecular surface electrostatic potential of these halogen-substituted BPAs. Table 3 and Figure 3 display the calculated results. The different halogen substitutions on BPA structure can impact the distribution and the size of positive and negative surface areas of the electron cloud. Figure 3 illustrates that the negative surface area of TBBPA (157.29 Å^2^) is larger than that of TCBPA (143.55 Å^2^). These differences arising from varying halogen substitutions lead to significant alterations in the molecular surface electrostatic potential, indicating that a more stable complex between HSA and TBBPA would be formed if the latter had a higher negative surface area [49]. This finding aligns with the observed increase in *K_a_* values.

### 2.3. Binding Site Analysis

HSA contains two major binding sites for ligand [50]. Among them, site I encompasses Trp and Tyr residues, while site II only contains Tyr residue. Synchronous fluorescence spectroscopy is a highly sensitive and selective technique that provides relevant insights into the interaction of ligands with aromatic amino acids. In the experiment, the monochromators were set for synchronous scans at fixed wavelength differences (∆λ) between the excitation wavelengths (λ_ex_) and the measured emission wavelengths (λ_em_). Synchronous fluorescence spectra of Tyr and Trp residues were obtained, with Δλ being 15 and 60 nm, respectively (Δλ = λ_em_ − λ_ex_). Hence, the effects of TBBPA/TCBPA on Trp and Tyr residues were explored by synchronous fluorescence spectroscopy. Upon the addition of TBBPA/TCBPA, a significant decline in the fluorescence intensity of Trp is observed, while the fluorescence intensity of Tyr experiences a slight decrease (Figure 4). These phenomena indicate that the binding site of HSA with TBBPA/TCBPA is close to Trp [51], and the microenvironment surrounding Trp is influenced by the presence of TBBPA/TCBPA. In summary, TBBPA/TCBPA enters site I and binds to HSA.

HSA possesses distinct hydrophobic cavities for ligand binding, and through site-specific competition experiments, we can identify the specific binding sites of ligands on HSA [10,52]. In this work, three representative site-specific competition probes were chosen to determine the specific binding sites of TBBPA/TCBPA with HSA [37]. Table 4 presents the results, showing that the presence or absence of ibuprofen or digoxin has little impact on the *K_a_* values of TBBPA/TCBPA with HSA (*p* > 0.05, n = 3). However, the *K_a_* values of TBBPA/TCBPA with HSA noticeably decrease upon the addition of phenylbutazone (*p* < 0.01, n = 3). This suggests competition between phenylbutazone and TBBPA/TCBPA for binding sites. Hence, it can be said that the primary binding site of TBBPA/TCBPA with HSA is site I [53], which aligns with the findings from synchronous fluorescence spectroscopy.

### 2.4. Conformational Changes Study

To investigate the conformational changes induced by TBBPA/TCBPA in HSA, both qualitative and quantitative analyses were conducted using 3D fluorescence spectra and CD spectra.

Three-dimensional fluorescence spectra allow for qualitative assessment of protein conformational changes [12,31]. Two characteristic peaks of HSA can be observed in 3D spectra (Figure 5). Peak 1 represents the properties of Trp and Tyr residues [54], and peak 2 reflects the fluorescence spectrum associated with the protein–peptide chain structure, which is closely related to the secondary structure of protein [55]. Table 5 presented the complete data of the 3D fluorescence spectra. In the TBBPA and TCBPA groups, the intensity of peak 1 significantly decreases, and the presence of ligands induced blue shift. This indicates the insertion of TBBPA/TCBPA into the binding sites of HSA, leading to changes in the microenvironment surrounding Trp and Tyr residues [56]. Moreover, the Stokes shift of peak 2 changes by 6 nm and 1 nm, respectively, while the intensity of peak 2 decreases to 575.5 and 692.6. These observations suggest that TBBPA/TCBPA induces changes in the secondary structure of HSA, and TBBPA exerts a stronger impact on HSA conformation compared to TCBPA.

CD spectroscopy can quantitatively analyze the specific changes of protein secondary structure, providing more particular information regarding the conformational changes of HSA upon interaction with TBBPA/TCBPA [10]. HSA displays two negative peaks at 208 nm (π → π*) and 220 nm (n → π*), representing the characteristic α-helical structure of HSA (Figure 6) [57]. The CD peak intensity at 220 nm of HSA undergoes significant changes in the presence of TBBPA/TCBPA (Figure 6), indicating alterations in the spatial conformation of HSA induced by TBBPA/TCBPA and modifications to HSA’s secondary structure [58]. Moreover, the effect of TBBPA on the CD spectrum of HSA is more pronounced compared to TCBPA, revealing a greater impact of TBBPA on HSA conformation.

Subsequently, the data were analyzed using the CONTINLL program [59]. The findings show that the α-helix content in free HSA is 51.4%, the β-turn content is 14.1%, the β-sheet content is 7.7%, and the random coil content is 26.9% (Table 6). Upon the addition of ligands, a similar change was observed in different groups. Specifically, the α-helix content slightly decreased from 51.4% to 48.2% and 50.4% upon interaction with TBBPA and TCBPA, respectively. These quantitative data demonstrate that the presence of TBBPA/TCBPA can cause changes in HSA conformation, with TBBPA exhibiting a more significant effect on HSA conformation compared to TCBPA. Furthermore, changes in protein structure may impact its physiological function [60]. Therefore, the conformational alterations in HSA caused by TBBPA/TCBPA may disrupt the bioactivity of HSA in the human body.

### 2.5. Molecular Docking 

Molecular docking was employed in this study to explore the binding modes of HSA with TBBPA/TCBPA, providing additional information at the molecular level [32,61]. TBBPA/TCBPA insert into the IIA subdomain of HSA (site I), which was confirmed by site competition experiments. As a result, the docking center was set at site I in this work. The conformation of the TBBPA/TCBPA–HSA complex was simulated by the AutoDock Vina program [62]. These data support the conclusion that TBBPA/TCBPA can spontaneously enter site I of HSA. The docking results revealed that the binding energies for TBBPA–HSA and TCBPA–HSA were −7.5 kcal/mol and −7.4 kcal/mol, respectively. This qualitatively indicates that the affinity of TBBPA for HSA is higher than that of TCBPA, which is consistent with the previous findings from the solution experiments.

Figure 7 illustrates the optimal docking models of HSA–TBBPA and HSA–TCBPA, where TBBPA/TCBPA is surrounded by numerous residues upon entering the HSA cavity. The π electrons on the phenyl rings of TBBPA and TCBPA are involved in pi–pi interactions with Trp214. This is probably what causes the fluorescence-quenching mechanism of TBBPA/TCBPA in HSA (Trp214 being a major contributor to HSA fluorescence intensity [63]). It is worth noting that TBBPA/TCBPA interacts with residues Lys199, Lys195, Ser202, Phe211, Arg218, Arg222, His242, Leu481, and Trp214. These results suggest that these amino acid residues may play a crucial role in the binding of BPA-like compounds to HSA. Moreover, the halogen substituents on TBBPA and TCBPA engage in pi–alkyl interactions with certain amino acid residues of HSA, indicating the presence of halogen substituents favors the formation of stable complexes between BPA-like compounds and proteins. According to the literature, the *K_a_* value for BPA–HSA is at the level of 10^3^ [23], while the K_a_ values obtained in this experiment are at the level of 10^5^. Therefore, these data partially support the aforementioned conclusions.

## 3. Materials and Methods

### 3.1. Instruments and Software

Steady-state fluorescence measurements (with a single-cell Peltier accessory temperature control system) were conducted by using a Cary Eclipse fluorescence spectrophotometer (Agilent Technologies, Palo Alto, CA, USA). pH measurements were performed using an EL20 precision pH meter (Mettler Toledo Inc., Schwerzenbach, Switzerland). UV-visible absorption data detection was performed using a GENESYS 50 UV-Visible spectrophotometer (Thermo Scientific, Carlsbad, CA, USA). Protein secondary structure analysis was carried out by using a J-1700 circular dichroism spectrometer (JASCO, Ishikawa-machi, Tokyo, Japan). AutoDock Vina 1.1.2 and Discovery Studio Client software 2018 were utilized for building the ligand (TBBPA/TCBPA)–receptor (HSA) models and analyzing their interactions [62]. The Multiwfn software 3.8 was employed to calculate the surface charge distribution of TBBPA and TCBPA [48].

### 3.2. Experimental Chemicals

TBBPA, TCBPA, and site-specific competitors (ibuprofen, digitonin, and phenylbutazone) were supplied by Shanghai Macklin Biochemical Co., Ltd., Shanghai, China. TBBPA and TCBPA were dissolved in absolute methanol to prepare stock solutions (1.0 × 10^−2^ M) and stored at −20 °C before use. HSA and Tris were obtained from Sigma-Aldrich, St. Louis, MI, USA and Shanghai Yuan Ye Biological Technology Co., Ltd., Shanghai, China, respectively. The stock solutions were diluted to appropriate concentrations according to experimental requirements. A Tris-HCl buffer solution containing 0.9% NaCl (pH 7.4) was used to imitate the physiological environment in the human body. All reagents used in this work were of analytical grade or higher.

### 3.3. Experimental Methods

#### 3.3.1. UV Absorption Experiment

The concentration of HSA was set at 4.0 × 10^−6^ M, while the concentration of the ligands (TBBPA/TCBPA) was varied as follows: [TBBPA] = 0, 1.0, 2.0, 3.0, 4.0 × 10^−5^ M; [TCBPA] = 0, 1.0, 2.0, 3.0, 4.0 × 10^−6^ M. UV absorption spectra of HSA were recorded in the range of 200–320 nm. The corresponding concentration solutions of the ligands were used as the blank solution, which were used to eliminate the effect of TBBPA/TCBPA on the UV-vis absorption spectra.

#### 3.3.2. Fluorescence–Thermodynamic Experiment

Prior to taking steady-state fluorescence measurements, the HSA and ligand mixture solution was given 2 min to equilibrate. At different temperatures (298, 303, and 310 K), fluorescence emission spectra of HSA were recorded in the range of 305–445 nm while varying the concentration of TBBPA/TCBPA. The concentration of HSA was fixed at 1.0 × 10^−6^ M. For TBBPA, it was sequentially added to achieve a concentration range from 0 to 5.0 × 10^−7^ M, with each change being 1.0 × 10^−7^ M. For TCBPA, it was sequentially added to achieve a concentration range from 0 to 7.5 × 10^−7^ M, with each change being 1.5 × 10^−7^ M.

To account for the ligand concentration in the study system, which was much lower than the 10^−5^ M level, the inner-filter effect of the ligand was neglected. The Stern–Volmer equation and double logarithmic equation (Equations (1)–(3)) were used to analyze the data of fluorescence spectra. Thermodynamic parameters were obtained using Equation (4) to determine the driving forces in the binding process.
(1)$$\frac{F_{0}}{F}=1+K_{sv}\left[Q\right]$$
(2)$$K_{q}=\frac{K_{sv}}{\tau_{0}}$$
(3)$$log\frac{F_{0}-F}{F}=logK_{a}+nlog\left[Q\right]$$
(4)$$∆G=∆H-T∆S=-RTLnK_{a}$$
where *F*_0_ and *F* are the fluorescence intensities of the study system with and without ligand, respectively; [*Q*] is the concentration of the ligand; *K_sv_* and *K_q_* are the Stern–Volmer quenching constant and the quenching rate constant of the biomolecule (M^−1^ s^−1^), respectively; *τ*_0_ (6.30 ns) is the fluorescence lifetime value of free HSA [64]; *K_a_* and n are the binding constant and the number of binding sites of the ligand–receptor, respectively; Δ*G*, Δ*H*, and Δ*S* are the changes in Gibbs free energy, enthalpy, and entropy during the ligand–receptor binding process, respectively. In addition, *R* and *T* are the gas constant at room temperature and the temperature in Kelvin, respectively.

#### 3.3.3. Synchronous Fluorescence Experiment

In this study, the wavelength intervals (Δ*λ* = *λ_em_* − *λ_ex_*) were fixed at 15 to gather fluorescence characteristic information from the Tyr residues, whereas Δ*λ* was fixed at 60 to gather fluorescence characteristic information from the Trp residues [65]. The data were recorded in the wavelength range of 245–315 nm.

#### 3.3.4. Site-Specific Competitive Experiment

The site-specific competitive experiment was conducted following the methodology described in the literature [37], using digitonin, ibuprofen, and phenylbutazone as site-specific probes. First, a 1:1 complex of the site-specific probe with HSA was prepared in the buffer solution. Then, TBBPA/TCBPA was added, and the influence of the site-specific probes on the binding ability of TBBPA/TCBPA with HSA was explored by measuring the changes in fluorescence intensity. This experiment was conducted under the same test conditions as the thermodynamic experiment, except at room temperature.

#### 3.3.5. Three-Dimensional (3D) Fluorescence Experiment

The concentration of the HSA solution was fixed at 1.0 × 10^−6^ M. The effect of TBBPA/TCBPA on the 3D fluorescence spectra of HSA was measured when the concentrations of ligands were 1.0 × 10^−6^ M. The instrumental parameters were then set as follows: the step sizes were 1 nm; the excitation and emission slit widths were set to 10 nm; excitation wavelength range and emission wavelength range were 200–300 nm and 295–445 nm, respectively. 

#### 3.3.6. Circular Dichroism (CD) Spectroscopy Experiment

CD spectra of HSA with and without TBBPA/TCBPA were measured in the range of 200–260 nm, and the CONTINLL program [59] was utilized to calculate the secondary structure content of HSA. The concentration of HSA and the ligands was set as 1.0 × 10^−6^ M. The instrumental parameters were set as follows: path length of 1 mm and scanning speed of 100 nm min^−1^.

#### 3.3.7. Molecular Docking Calculation

The 1H9Z crystal structure in the RCSB Protein Data Bank (http://www.rcsb.org/pdb accessed on 15 May 2023) was used as the three-dimensional structure of HSA. In this structure, the protonation states of residues were at pH 7.0 [66]. All water molecules and ligands in the 1H9Z crystal structure were removed, and Kollman charges and polar hydrogens were added to residues by using MGLTools-1.5.6 software. The structures of TBBPA and TCBPA were built using Chem Office and optimized under MMFF94 force field conditions, and then converted into pdbqt format. The optimal configurations of the TBBPA/TCBPA–HSA complexes were constructed using AutoDock Vina 1.1.2 according to the official tutorial (https://vina.scripps.edu/tutorial/ accessed on 15 May 2023). Finally, the results were showed in the form of pictures with the help of PyMOL-2.5 and Discovery Studio Visualizer 2018 Client. Additionally, the ligand conformations constructed and optimized using Chem Office were used to calculate the surface charge and distribution of the ligands using Multiwfn software 3.8 [48].

## 4. Conclusions

This study performed a combination of multispectral, theoretical calculation, and molecular docking to generate credible and comprehensive experimental data. The investigation focused on the interaction mechanisms between HSA and TBBPA/TCBPA, as well as the impact of different halogen substitutions on their binding process.

The results reveal that TBBPA/TCBPA can effectively insert into site I of HSA, forming stable binary complexes. Hydrophobic interactions primarily drive the formation of the TBBPA–HSA complex, whereas the TCBPA–HSA complex is derived by hydrogen bonding and van der Waals forces. The binding constant of TBBPA–HSA exceeds that of TCBPA–HSA by threefold. The presence of TBBPA/TCBPA induces conformational changes and alters the microenvironment of HSA, with TBBPA exhibiting a stronger influence on the secondary structure compared to TCBPA. These structural modifications in HSA imply potential interference with its biological activity, thus highlighting the associated health hazards for humans.

Furthermore, the inclusion of halogen substituents facilitates the stability of the BPA-like compounds–HSA complex. The differential impact of various halogen substitutions on the negative electrostatic surface area of BPA emerges as a significant factor relative to the higher binding affinity of TBBPA with HSA when compared to TCBPA. Notably, the stronger binding between TBBPA and HSA results in an extended half-life and slower metabolism within the body, warranting additional attention towards the long-lasting effects of TBBPA on humans. Thus, this study enhances our understanding of the binding mechanisms between halogenated BPA compounds and HSA, providing valuable theoretical support for managing human health risks.

## Figures and Tables

**Figure 1 ijms-24-13281-f001:**
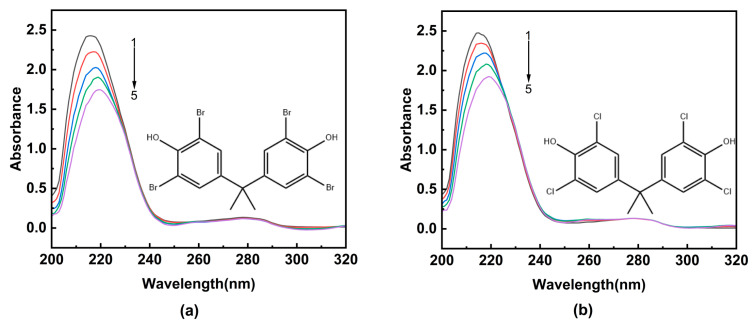
The UV-visible absorption spectra of HSA in the presence of ligands. (**a**) TBBPA; (**b**) TCBPA. [HSA] was 4.0 × 10^−6^ M, and [TBBPA] and [TCBPA] were varied as indicated: [TBBPA] = 0, 1.0, 2.0, 3.0, 4.0 × 10^−5^ M from curves 1 → 5; [TCBPA] = 0, 1.0, 2.0, 3.0, 4.0 × 10^−6^ M from curves 1 → 5.

**Figure 2 ijms-24-13281-f002:**
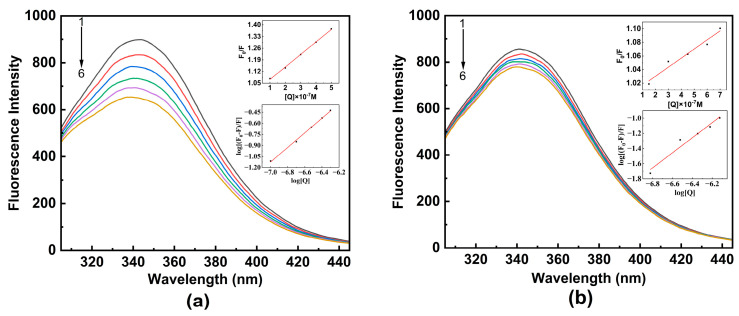
The fluorescence spectra of HSA with ligands at 298 K. (**a**) TBBPA; (**b**) TCBPA. The insets of the figure show the Stern–Volmer and double logarithmic equation curves for TBBPA–HSA and TCBPA–HSA. [HSA] = 1.0 × 10^−6^ M; [TBBPA] = 0, 1.0, 2.0, 3.0, 4.0, 5.0 × 10^−6^ M; [TCBPA] = 0, 1.5, 3.0, 4.5, 6.0, 7.5 × 10^−6^ M.

**Figure 3 ijms-24-13281-f003:**
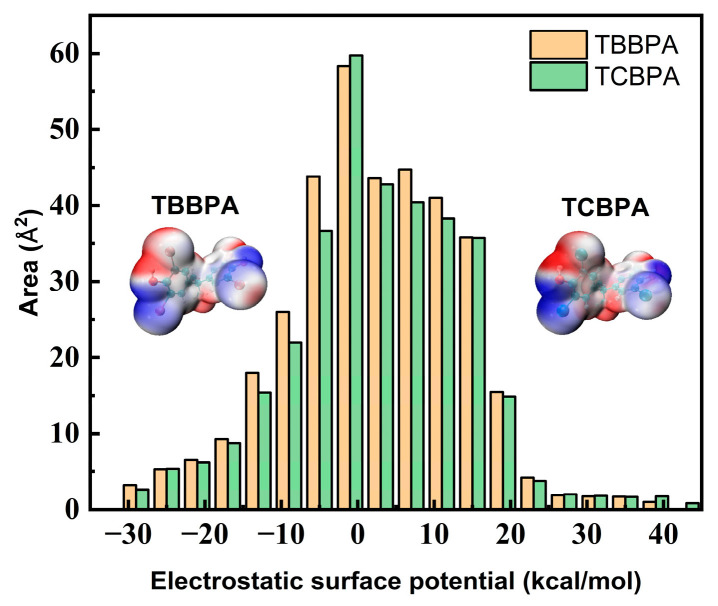
Surface charge and distribution of TBBPA and TCBPA.

**Figure 4 ijms-24-13281-f004:**
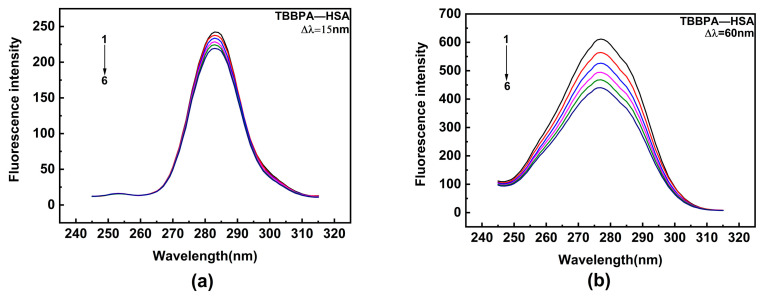
Synchronous fluorescence spectra. (**a**,**b**) TBBPA–HSA; (**c**,**d**) TCBPA-HSA. Here, [HSA] = 1.0 × 10^−6^ M; [TCBPA] = 0, 1.5, 3.0, 4.5, 6.0, 7.5 × 10^−6^ M; [TBBPA] = 0, 1.0, 2.0, 3.0, 4.0, 5.0 × 10^−6^ M.

**Figure 5 ijms-24-13281-f005:**
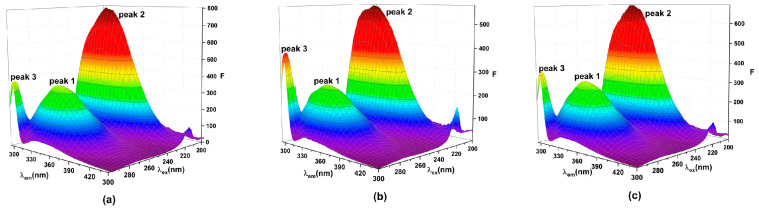
Three-dimensional fluorescence spectra. (**a**) HSA; (**b**) TBBPA-HSA; (**c**) TCBPA-HSA. [HSA] = [TBBPA] = [TCBPA] = 1.0 × 10^−6^ M.

**Figure 6 ijms-24-13281-f006:**
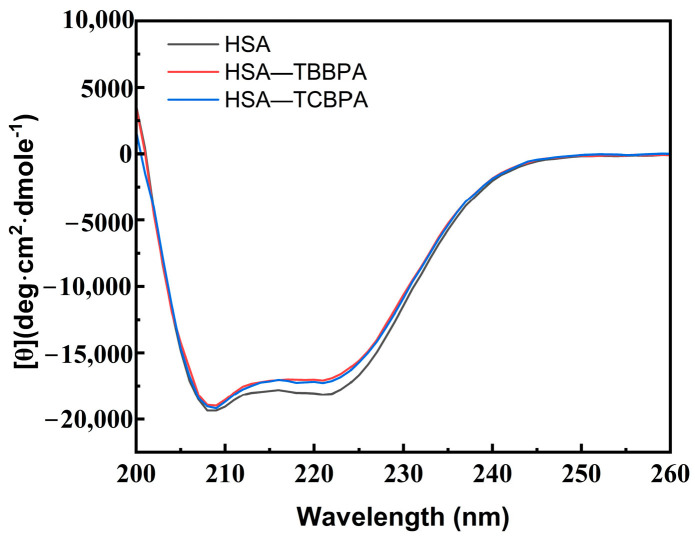
CD spectra of HSA with and without TBBPA/TCBPA. [HSA] = [TBBPA] = [TCBPA] = 1.0 × 10^−6^ M.

**Figure 7 ijms-24-13281-f007:**
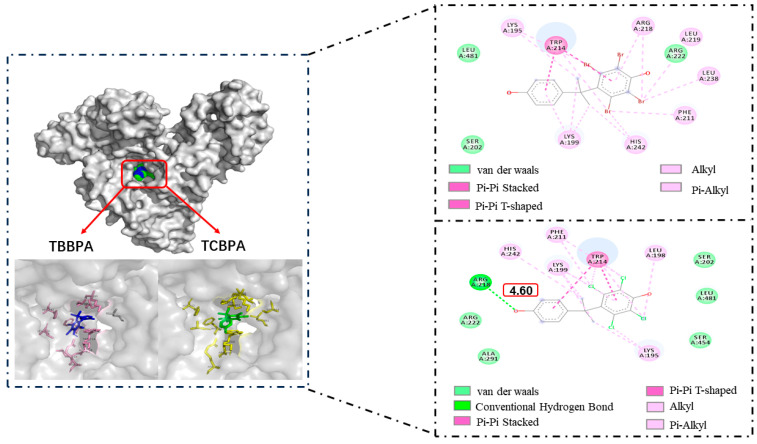
Molecular docking of TBBPA/TCBPA–HSA.

**Table 1 ijms-24-13281-t001:** Stern–Volmer quenching parameters.

	*T* (K)	*K_sv_* (10^5^ M^−1^)	*K_q_* (10^14^ M^−1^·s^−1^)	*R* ^2^	n
**TBBPA–HSA**	298	7.528	1.210	0.999	0.997
	303	7.534	1.211	0.998	0.995
	310	7.627	1.226	0.997	0.997
**TCBPA–HSA**	298	1.328	0.2135	0.995	1.003
	303	1.296	0.2083	0.987	1.001
	310	1.289	0.2073	0.984	1.003

**Table 2 ijms-24-13281-t002:** Thermodynamic parameters of present work.

	*T* (K)	*K_a_* (10^5^ M^−1^)	*R* ^2^	n	∆*H* (kJ·moL^−1^)	∆*S* (J·moL^−1^·K^−1^)	∆*G* (kJ·moL^−1^)
**TBBPA–HSA**	298	6.245	0.999	0.9887	19.19	175.4	−33.08
	303	7.219	0.999	0.9987			−33.95
	310	8.443	0.998	1.008			−35.18
**TCBPA–HSA**	298	2.040	0.996	1.029	−55.35	−84.86	−30.07
	303	1.101	0.993	0.9873			−29.64
	310	0.8391	0.995	0.9664			−29.05

**Table 3 ijms-24-13281-t003:** Detailed data of surface charge for TBBPA and TCBPA.

Compounds (298 K)	*K_a_* (10^5^ M^−1^)	MPI (K·moL^−1^)	A_t_ (Å^2^)	A_p_ (Å^2^)	A_n_ (Å^2^)	V_min_ (KJ·moL^−1^)	V_max_ (KJ·moL^−1^)
**TBBPA**	6.245	9.048	361.62	204.33	157.29	−29.73	39.75
**TCBPA**	2.04	9.158	340.64	197.09	143.55	−29.22	43.3

**Table 4 ijms-24-13281-t004:** The *K_a_* values of TBBPA/TCBPA–HSA with competing probes.

	[Ligands]/ [Site Marker]	TBBPA–HSA *K_a_* (10^5^ M^−1^)	TCBPA–HSA *K_a_* (10^5^ M^−1^)
**Control group**	1:0	13.63 ± 0.11	7.32 ± 0.08
**Phenylbutazone**	1:1	8.14 ± 0.14	3.19 ± 0.18
**Ibuprofen**	1:1	13.52 ± 0.14	7.60 ± 0.15
**Digoxigenin**	1:1	13.45 ± 0.25	7.83 ± 0.35

**Table 5 ijms-24-13281-t005:** Relevant data from the three-dimensional fluorescence spectra.

	Peak 1			Peak 2		
	λ_ex_/λ_em_ (nm/nm)	∆λ (nm)	*F*	λ_ex_/λ_em_ (nm/nm)	∆λ (nm)	*F*
**HSA**	279/341	62	368.7	225/334	109	801.7
**TBBPA–** **HSA**	279/338	59	254.0	225/328	103	575.5
**TCBPA–HSA**	279/340	61	323.4	226/334	108	692.6

**Table 6 ijms-24-13281-t006:** Relevant data of CD spectra.

	α-Helix/%	β-Sheet/%	β-Turn/%	Random Coil/%
**HSA**	51.4	7.7	14.1	26.9
**TBBPA–HSA**	48.2	7.3	14.7	29.8
**TCBPA–HSA**	50.4	8.2	14.2	27.2

## Data Availability

Data will be made available on request.

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
