# Peer review of "Impacts of Halogen Substitutions on Bisphenol A Compounds Interaction with Human Serum Albumin: Exploring from Spectroscopic Techniques and Computer Simulations"

_ijms, 2023, doi:10.3390/ijms241713281_

Round 1
Reviewer 1 Report
Zhang et al. report an interaction study of human serum albumin and two halogen-substituted bisphenol A derivatives. They used various spectroscopic techniques (UV-vis absorption, fluorescence, CD) supplemented by calculations. They observed and explained the difference in affinity of TBBPA and TCBPA for HSA. I consider the work well designed, the experiments well performed, the paper well written. The conclusions are supported by the data obtained and presented. Figures and tables are appropriate and clear (designed). I believe that the paper can potentially be of interest to a relatively wider audience, yet it deserves minor corrections. After that, I would like to recommend the paper for publication in IJMS.
1) p.2, line 87: Autodock VNIA should be Autodock VINA. Add also references to all programs.
2) p.3, section 2.3.1: Did you perform any baseline correction? How the background signal was treated?
3) p.4, section 2.3.6: Full details of the experimental approach, including concentrations used, should be given here. The reference of the CONTINLL program is missing. I guess, the path length of 1 nm is a mistake.
4) p.4, line 159, more info about preprocessing the pdb structure should be also given here (protonation states of residues according to a chosen pH, deleting or keeping of residual solvent molecules, etc.). Was any continuous solvent model employed during the optimization at the MMFF94 level?
5) Figure 1.: Figure shows five spectra labeled as 1-5, but in the caption, there is only mentioned the concentration scale. It is not clear, if 1 = 0.0 or 4.0 x 10^-5 (10^-6).
6) Figure 2. Inset panels are hardly visible.
7) p.7, line 263: there is a sentence about the positive delta S indicating dominant hydrophobic interactions. This deserves some explanation and/or some references.
8) Figure 4. the delta lambda value is not explained (similarly in Table 5). I think, lambda(ex) and lamda(em) could be also explained here again for readers convenience.
9) p.10, line 326: spectrc should be spectrum/spectra.
10) p.10, line 352 and below: I think, changes in the secondary structure derived from the CD spectra are relatively small. I would guess, at the limit of experimental and the computational model accuracy. Therefore, I would suggest to recalculate the composition using also other models (e.g. K2D3 - http://cbdm-01.zdv.uni-mainz.de/~andrade/k2d3/; Bestsel - https://bestsel.elte.hu/index.php, etc.)
and compare the results. Nevertheless, I would soften the statement and just said that the interaction with the ligand causes a slight decrease of the helical content (or something like this).
11) p.11, line 362: When said, any further investigations are warranted, you could try to predict to ADME/Toxicity properties (e.g. http://www.swissadme.ch)
or some metabolism predictions (e.g. http://eawag-bbd.ethz.ch/predict/, https://biotransformer.ca/new)
to support such statement. Or delete it.
12) p.11, line 378: Do binding energies of -7.5 and -7.4 represent calculated values provided by the docking? If so, then it should be emphasized that docking is usually used to qualitatively sort the ligands and not to provide absolute numbers.
Reviewer 2 Report
The manuscript ijms-2560384 presents a study about the interaction of halogen-substituted bisphenol A compounds with human serum albumin.
Introduction. Because the authors describe the binding of TBBPA/TCBPA into the IIA subdomain of HSA, it would be interesting to add some general details about the structure of HSA and the relevance of the IIA subdomain.
Materials and methods. The authors must add from the very beginning the references for the methods applied - page 2, page 4 L158,etc.
L87 - Is it correct "Autodock VNIA"?
page 6 L235 Please add the reference for "In vivo, binding constants for drug-protein interactions typically fall within the range of 104 ~ 106 M-1 . Binding constants in the range of 104 M-1 and 105 ~ 106 M-1 generally signify moderate and strong binding strengths, respectively, between the ligand and HSA"
Figure 7 is too small.
